# In-Volume Glass Modification Using a Femtosecond Laser: Comparison Between Repetitive Single-Pulse, MHz Burst, and GHz Burst Regimes

**DOI:** 10.3390/ma18010078

**Published:** 2024-12-27

**Authors:** Manon Lafargue, Théo Guilberteau, Pierre Balage, Bastien Gavory, John Lopez, Inka Manek-Hönninger

**Affiliations:** 1Université de Bordeaux-CNRS-CEA, CELIA UMR5107, 33405 Talence, France; 2Amplitude, Cité de la Photonique, 11 Avenue de Canteranne, 33600 Pessac, France; 3ALPhANOV, Rue François Mitterrand, 33400 Talence, France

**Keywords:** ultrafast laser processing, femtosecond GHz bursts, laser–material interaction, transparent welding, glass

## Abstract

In this study, we report, for the first time, to the best of our knowledge, on in-volume glass modifications produced by GHz bursts of femtosecond pulses. We compare three distinct methods of energy deposition in glass, i.e., the single-pulse, MHz burst, and GHz burst regimes, and evaluate the resulting modifications. Specifically, we investigate in-volume modifications produced by each regime under varying parameters such as the pulse/burst energy, the scanning velocity, and the number of pulses in the burst, with the aim of establishing welding process windows for both sodalime and fused silica.

## 1. Introduction

Recent advancements in ultrashort pulsed (USP), typically femtosecond and picosecond, laser glass welding have paved the way for innovative applications [1,2,3,4,5]. Indeed, localized heating minimizes thermal damage, making it ideal for brittle and heat-sensitive glass components, particularly in temperature-sensitive technologies such as the integration of electronic or optoelectronic devices such as MEMS or in the fields of optics, microfluidics, and medical devices [6,7]. The laser welding process involves nonlinear absorption [8], leading to the formation of a microscopic melt pool that typically reaches only a few hundred micrometers in size. This enables rapid cooling and efficient heat dissipation, thereby minimizing thermal effects outside the focusing zone and allowing for high precision [9].

Despite these advantages, ultrafast laser welding faces several challenges. The small melt pool size limits the ability to bridge gaps [10] and requires higher precision in the positioning of the laser spot with respect to the glass–glass interface. Current studies are focused on optimizing the welding process by exploring different laser regimes and investigating how materials respond under these conditions, such as the MHz burst mode [11]. A high breaking resistance weld was achieved in the study by Zimmermann et al., where using bursts helped to reduce stress around the laser-induced weld seams [12].

The GHz burst mode, a relatively new regime, has gained increasing attention in recent years and is becoming more prominent in various glass processing applications [13,14,15,16]. Its unique capability to gradually heat materials through specific energy deposition mechanisms could enhance the melt pool size, facilitating a more robust and efficient welding process. Spreading the incoming pulse energy into a burst of several tens of femtosecond pulses contributes to reducing nonlinear propagation effects and energy deposition upstream of the focus point, so one can expect a different bulk modification compared to repetitive single pulses.

In this contribution, we present a comparative study of bulk modification morphologies in the repetitive single-pulse, MHz burst, and GHz burst regimes on sodalime and fused silica. The aim of this study was to establish the process windows, in terms of pulse/burst energy and scanning velocity, for the three regimes in order to gain a better understanding of the process. The results will be discussed in terms of the quality of the welding seam (crack-free, transparency, and uniformity).

## 2. Materials and Methods

### 2.1. Laser System and Machining Workstation

This experiment was conducted using an Yb-doped femtosecond laser, based on the commercial Tangor 100 system from Amplitude (Pessac, France). The laser provides pulses at a wavelength of 1030 nm with a pulse duration of 500 fs and a maximum average output power of 100 W. This laser system can operate in the single-pulse mode, the MHz burst mode with up to 32 pulses per burst at 40 MHz, or the GHz burst mode with 50 pulses per burst at 1.28 GHz. This flexible system, fully detailed in reference [17], allows for a robust comparison between all three regimes.

The burst repetition rate is adjustable from 1 Hz to 2 MHz, and the pulse energy can be varied using a half-wave plate followed by a polarizing beam splitter cube (see Figure 1a). The rotation of the half-wave plate is automated and controlled through DMCpro software v1.5.x (Direct Machining Control, Vilnius, Lithuania). A calibration was performed between the selected angle and the power value measured with a pyroelectric power meter (Gentec, Quebec city, QC, Canada, UP19K-50F-W5-D0) underneath the scanner. A beam expander was set in order to obtain a 12 mm collimated beam before entering the scanner and focusing lens. As a result, with a telecentric lens (f = 30 mm, NA = 0.2), a measured spot diameter of 6.3 μm at 1/$e^{2}$ was obtained using a beam analyzer (DataRay, Redding, CA, USA, DSAT-WinCamD-XHR, pixel size 3.3 μm) equipped with a 10× homemade calibrated zoom. The chosen numerical aperture offers an optimal compromise between tight focusing and scanning speed. It is high enough to ensure a small focal spot while still allowing for flexibility and efficient movement of the spot in the target using a Galvo scanner (Scanlab, Puchheim, Germany, IntelliScan III-14). A higher NA would reduce the scanning area and require slower speeds to maintain the focus; in our case, this specific lens provides the needed focus quality without compromising scanning velocity, thus enabling the effective investigation of velocity effects on the seam morphology.

The Galvo scanner and the off-axis front camera (Basler (Anyang-si, Republic of Korea) acA2000-50gm, resolution of 2048 × 1088, pixel size 5.5 μm × 5.5 μm, 50 fps, global shutter) were both fixed on a motorized Z-axis stage (Alio Industries, Arvada, CO, USA, AI-LM-10000-I-PLT-LP) for precise positioning of the focal plane within the sample. The sample was placed on XY monolithic motorized stages (Alio Industries, AI-LM-20000-XY-I-LP) to ensure proper positioning under the scanner head. The station was also equipped with a side-view real-time imaging system, including a green diode emitting at 523 nm and a Basler CMOS camera (Basler acA1440-220um, resolution of 1440 × 1080, pixel size 3.45 μm × 3.45 μm, 227 fps, global shutter) coupled with a long-distance microscope (InfiniMax KX (Centennial, CO, USA) with MX4 objective). A 520 nm bandpass filter was added upstream of the camera to visualize the samples without being blinded by the laser processing wavelength. The translation stages, laser gate, and power modulator unit were also controlled by DMCpro software. The workstation had a granite base and gantry, ensuring the stability and repeatability of the experiments.

For this study, we selected two different types of glass, namely sodalime, which is highly accessible, low-cost, and widely used in many application fields such as microelectronics and the automotive industry, and fused silica, which is primarily used for optical components. We used commercial sodalime (RS BPB016) and fused silica samples from Altechna (Vilnius, Lithuania) (JGS1 22 × 30 × 1; 80-50 S-D). The latter samples were polished on both sides to facilitate side observation. The glass thickness was 1 mm for both materials.

### 2.2. Experimental Protocol

The principle of the experiment to produce the bulk modification is schematically depicted in Figure 1b. Once the sample was correctly set vertically under the Galvo scanner and the focusing lens, the focal plane was set at 500 μm below the surface, taking into account the refractive index of the glass sample. Then, a line of 0.6 mm was drawn along the X-axis using the scanner at the desired velocity and with the different laser parameters. It is important to have a line shorter than the thickness of the material to stay within the volume and avoid any ablation at the air–glass interface on both vertical sidewalls of the glass sample. The side-view camera allowed for direct observation of the modification created in the glass, which will be described in the paragraph below. The sample was then translated by 300 μm along the Y-axis, and another line was drawn for the next set of parameters.

In this study, we chose fixed repetition rates of 100 kHz and 400 kHz to be able to compare low and high repetition rates. This approach allowed us to observe the response of the material both without and with heat accumulation, which occurred at repetition rates above 200 kHz due to the pulse period being shorter than the time required for heat diffusion [18]. The scanning velocity of the scanner was set at different values ranging from 1 mm/s to 1 m/s. In order to observe the influence of the temporal beam shaping, we chose a comparable energy range, from 0.1 to 200 μJ per pulse or burst, for the three regimes. This means that a single pulse carried as much energy as a whole burst in this study.

## 3. Results and Discussion

### 3.1. Classification and Analysis of the Induced Modifications

A post-mortem observation of the sample under a microscope (MF-B1010D Mitutoyo) equipped with a 10× objective was carried out. Based on lateral observation, bulk modifications were characterized using specific criteria to determine their classification into one of the six categories listed below. Each class of modification is described below and depicted in Figure 2.

No interaction: no visible permanent bulk modification was observed.Elongated bulk modification: Due to the high intensity focused within the dielectric, nonlinear absorption occurs. Initially, a few electrons in the valence band are promoted to the conduction band through photoionization. Subsequently, hot free electrons in the conduction band may further absorb energy by linear absorption from the incoming laser pulse, increasing their temperature and kinetic energy; this phenomenon is called electron heating. When the electron kinetic energy is of the order of the bandgap energy, these electrons can collide with valence electrons, leading to impact ionization. Then, at a timescale of a few hundred femtoseconds [19,20], hot free electrons transfer their heat to the glass lattice owing to electron–photon coupling [21]. As a result, the laser energy is efficiently absorbed near the focal region. Then, high temperature and high pressure produce physical effects, such as thermal dilatation, melting, or element migration [22], resulting in local index change [23], density variation [24], or residual stress.Teardrop: The modified region has a teardrop shape with a dual structure. The inner structure corresponds to the initial seed electrons produced through nonlinear ionization. When the pulse-to-pulse delay is shorter than the mean relaxation time (in the order of magnitude of 10 μs for fused silica and sodalime), there is heat accumulation between successive pulses at a high repetition rate (above 100 kHz), causing the melted region to extend beyond the focal volume [19,25,26], producing the outer structure. The observed teardrop shape is explained by a shift in the main absorption position toward the focusing lens, resulting in an outer elliptical profile.Void: In addition to the heat-affected zone (HAZ), bubbles can also appear, though this phenomenon is not yet fully understood. The first description was provided by Cvecek et al., suggesting that gas bubble formation may originate from the decomposition of SiO_2_, cavitation mechanisms, or rapid quenching [27]. Multiple tiny bubbles appear first within the heat-affected zone, and then thanks to the low viscosity of glass under high temperature, they can migrate and aggregate into micro-bubbles at the top of the teardrop bulk modification. According to the literature, bubbles can appear at 2950 °C [27] in fused silica and at 1300 °C [28] in sodalime.Crack: Material fractures occur. Cracks can be formed due to internal stress caused by volume expansion.Crack at departure: At the beginning of the welding seam, the glass is cold and brittle, so the rapid heating and cooling due to laser irradiation can result in crack formation. Then, after a few tens of microseconds, the high repetition rate induces a rise in temperature, resulting in warmer and more ductile glass. A self-healing process comes into play and the cracks disappear along the seam [9,26].

An additional top-view observation was used to determine whether the formed seam was a stable or an unstable bulk modification.

Based on this experiment, we assumed that the optimum bulk modification for transparent glass welding should be a large, crack-free, void-free (Figure 2b), stable, and uniform seam. Moreover, the large melted volume, induced by heat accumulation and resulting in a teardrop shape, seemed to be more convenient to enlarge the contact area at the interface and create a strong bond between the two samples, compared to a thin elongated bulk modification in which the energy deposition occurred in a smaller volume. We hypothesized that a stable seam would be preferable over an unstable one in order to ensure the regularity and homogeneity of the weld.

### 3.2. Sodalime

In this section, we provide a thorough study of the bulk modification shapes obtained in sodalime with all three regimes. We investigated the overall morphology of the resulting in-volume modification, as mentioned above, for a wide range of burst/pulse energies and scanning velocities. Through this, we were able to determine the process windows for all three regimes using a similar protocol.

#### 3.2.1. Single-Pulse Regime

The results and the process windows are depicted in Figure 3a for a repetition rate of 100 kHz, and in Figure 3b for 400 kHz. For the sake of clarity, both axes are displayed using base-10 logarithmic scaling. The black crosses represent the sets of parameters leading to a bulk modification, whereas the white circles correspond to the tested experimental points where no visible modification was observed. Cracks are indicated by full black circles (dots). The half-filled circle indicates the formation of an unstable seam or cracks at the departure, depending on their orientation. The combination of both is indicated by the same symbol with three black quadrants. Additionally, to identify the types of modifications, the background color is set according to the corresponding modification as defined in Section 3.1.

As expected, a too-low energy deposition did not generate any bulk modification. At 100 kHz, the threshold energy to produce a bulk modification was 0.2 μJ/pulse at 1 mm/s. This threshold increased up to 0.5 μJ/pulse for scanning velocities ranging from 100 mm/s to 1 m/s. A heat-affected zone was observed at 3 μJ/pulse at 1 mm/s and at 14 μJ/pulse at 100 mm/s.

At 400 kHz, the behavior was more or less the same as that at low pulse energy, and it was possible to obtain a larger heat-affected zone [18]. Due to heat accumulation, a teardrop modification appeared at lower pulse energy compared to the 100 kHz repetition rate in the single-pulse regime. Indeed, the minimum pulse energy to achieve teardrop formation was 1.2 μJ/pulse at 1 mm/s, 3.5 μJ/pulse at 100 mm/s and 43 μJ/pulse at 1 m/s. This result is very promising for industrial applications in terms of yield. However, heat accumulation can induce (or provoke) the detrimental effects for (or at) high energies, such as crack formation (above 38 μJ/pulse at 100 mm/s) or an unstable seam at low scanning velocities (above 14 μJ/pulse at 100 mm/s).

#### 3.2.2. MHz Burst Regime

In this subsection, we use MHz bursts with 4 or 16 pulses per burst (ppb). The pulse repetition rate within the burst was fixed at 40 MHz, while the burst repetition rate was set to 100 kHz or 400 kHz. Note that within the burst, the pulse-to-pulse delay was 25 ns, which is highly sufficient for heat accumulation. The different plots in Figure 4 demonstrate a more complex behavior compared to the single-pulse regime.

Indeed, splitting the high-energy single pulse into a train of lower-intensity pulses decreased the detrimental nonlinear propagation effects upstream of the focus, such as the Kerr effect or intensity clamping; meanwhile, it enhanced the deposited energy density at the focus [29]. This latter phenomenon significantly increased the risk of crack or void formation in the bulk modification. It also dramatically increased the risk of an unstable welding seam. Furthermore, increasing the number of pulses in the burst (from 4 ppb to 16 ppb) decreased the chance of obtaining an HAZ without any voids or cracks and diminished the welding process window. These negative effects were even more visible at 400 kHz. In Figure 4b,c, we observe that voids appear at lower burst energy compared to cracks. Therefore, transparent welding in the MHz burst mode is challenging, as the parameter range for achieving a defect-free teardrop shape is narrow. However, it is still possible to use the elongated modification (blue zone) for welding.

#### 3.2.3. GHz Burst Regime

The GHz burst tests were conducted with the same protocol, using a burst length of 50 pulses per burst (ppb). The burst shape is flat, meaning the energy is almost equally distributed across all pulses [13]. The modifications are presented in Figure 5. As before, we maintained the operating repetition rates of 100 kHz and 400 kHz. The pulse repetition rate within the burst was 1.28 GHz, with each pulse separated by 0.8 ns.

In comparison to repetitive single pulses or MHz bursts, the pulse energy required to produce a bulk modification was 10 to 20 times less with GHz bursts and reached 5 to 8 μJ. Below this range, there was no modification, whereas above it, we observed elongated modifications, followed by cracks.

However, as shown in Figure 5, varying the scanning velocity from 1 mm/s to 300 mm/s had a slight influence on the bulk modification morphology. Cracks were predominant compared to teardrops or voids, which can be explained by the energy deposition behavior in the GHz burst regime. Unlike other regimes, where the energy distribution is more diffused, the thermal energy deposition in burst regimes is concentrated in the focal plane [19,29]. Based on this study, this behavior is accentuated in the GHz burst regime. Indeed, in this regime, the very confined local heat accumulation in the laser focus induced a strong radial temperature gradient, provoking a radial dilatation, which might lead to a high local shear stress, thus forming a crack [30] (see Figure 6b) or a more pronounced vertical crack along the modification at the focal plane (Figure 6d).

As for MHz bursts, transparent welding with GHz bursts was only possible through primary absorption, not HAZ. However, the effect of the burst on the deposited energy density was more visible with GHz bursts, and a slight variation of 20–30% in the burst energy could compromise the feasibility of the process itself. In contrast, in the MHz burst regime, a variation of one to two orders of magnitude was tolerated by the process, and in the single-pulse regime, the process even allowed for variations exceeding two orders of magnitude of the pulse energy. Therefore, the process with GHz bursts presented considerably less robustness than the two other regimes.

In summary, this comprehensive study of bulk modifications in sodalime glass shows that the way the energy is deposited into the material significantly impacts the resulting modifications. The single-pulse mode is more suitable for producing modifications without any cracks and for creating larger melted zones. The repetition rate plays a major role in the expansion of the heat-affected zone. The use of bursts increases the deposited energy density near the focus and thus increases the risk of defects such as voids or cracks, with a detrimental effect on the robustness of the process.

Nevertheless, according to [31], increasing the numerical aperture (NA) can introduce some geometrical aberration, spread out the energy deposition, and reduce the possibility of void or crack formation. However, it may also decrease the size of the modification. This study was conducted with a fixed NA of 0.2, and the results may vary slightly for other NA values. A balance should be found, as studies have shown that higher fracture resistance in welding can be achieved using bursts rather than single pulses.

### 3.3. Fused Silica

The same experimental protocol was applied in fused silica, which has a higher bandgap, a higher thermal conductivity and a higher softening temperature compared to sodalime. The values are listed in Table 1.

Since fused silica has a higher thermal conductivity, we chose to focus the experimental trials at a high repetition rate (400 kHz) for the MHz and GHz burst regimes in order to enhance heat accumulation near the focus. From a practical point of view, the major difference between fused silica and sodalime was that the teardrop modifications in fused silica were always accompanied by voids, as shown in Figure 7a. This phenomenon was already reported in [27,36] and in [37]. The origin of these bubbles is still not fully understood, but by adjusting the processing parameters (burst/pulse energy and scanning velocity), it is possible to control their aggregation in a self-organized pattern of micro-bubbles, as shown in Figure 7c. We assumed that this different behavior between fused silica and sodalime could be explained by the difference in their thermal conductivity and thermal expansion coefficient. Indeed, on the one hand, the lower thermal conductivity of sodalime increased the thermal gradient in the focal region. On the other hand, its much higher thermal expansion coefficient contributed to enhancing the shear stress that generated cracks before degassing occurred.

As previously discussed, there is no modification, exhibiting only a HAZ, since voids are always present. Compared to sodalime, the bulk modifications in fused silica are more elongated and thinner. This can be explained by the differences in the bandgap energy and thermal conductivity.

The minimum energy to produce a modification was 1.5 μJ for a repetitive single pulse, 3.5 μJ per burst for a MHz burst with 4 ppb, 10 μJ per burst for a burst of 16 ppb, and 10 μJ per burst for a 50 ppb GHz burst. This observation demonstrates that there is a synergetic effect between pulses within the burst. In the case of single pulses at low frequency and low speed (Figure 8a,b), we observed either primary absorption or voids. With pulse energies below 15 μJ, no heat accumulation occurred; however, once this threshold was exceeded, teardrop-shaped modifications with voids were formed. Note that we were unable to characterize bulk modifications at 1 mm/s using more than 40 μJ per pulse, as the size of the teardrop exceeded the thickness of the sample. The graph also highlights the effect of increasing speed: at 500 mm/s, more energy was required to achieve similar modifications compared to 100 mm/s.

This behavior was similar at higher frequencies, except that the energy level required to produce a HAZ with voids was reduced due to heat accumulation, as observed in sodalime. At low velocity and high energy, a crack appeared after the laser irradiation. It is interesting to observe another region of cracks at departure at high velocities. These crack morphologies differ; in the first case, they are caused by thermal expansion, while in the second case, these points indicate the transition from cold and brittle glass to hot and ductile glass.

In the MHz burst regime, primary absorption shapes rarely appeared. For shorter bursts (Figure 8c), void formation began around 4 μJ/burst, except at 500 mm/s, at which point only cracks were present. Using high energy at high velocities led to the formation of cracks. Below 30 μJ, a crack could occur at departure, but it was contained and could heal. As the energy increased, the intensity at the focal plane became too high for the material, resulting in cracks along the entire seam.

Increasing the number of pulses per burst increased the energy threshold for modification (around 10 μJ). The results for MHz bursts with 16 ppb and GHz bursts were very similar. All modifications consist of cracks, as shown in Figure 8d,e. The crack is located at the tip of the teardrop, where the focal plane is located (see Figure 9a). In the top-view photograph, we can observe the cracks extending along the entire seam (see Figure 9b).

Preliminary results suggest that the use of the GHz burst regime for glass–glass transparent welding may be limited. Further studies are needed to explore whether varying the burst duration or shape or the numerical aperture can expand the processing window.

Thanks to the processing window depicted in Figure 8b, we were able to select the appropriate energy and speed to weld two plates of fused silica. A velocity of 5 mm/s was used, with single pulses of 30.8 μJ energy at a repetition rate of 400 kHz. This point lies within the orange zone, indicating that the modification produced has a teardrop shape with a void. The plates were positioned in a homemade clamping tool, providing circular pressure without any surface preparation. A microscope objective (Mitutoyo NIR APO 10×) with an effective numerical aperture of 0.21 was used to focus the laser beam, resulting in a spot diameter of 5.8 μm at $1/e^{2}$. The off-axis camera allowed for visualizing and setting the position of the laser focus at the interface of the glass samples. A spiral pattern was applied to produce a continuous seam with a diameter of 10 mm and a distance of 50 μm between turns. A photograph of the resulting welding over this large diameter is shown in Figure 10a. Due to the presence of voids, the welding zone is not transparent, as illustrated in Figure 10b. This constitutes a clear limitation for applications in the welding of optical components, where fused silica is used as an optical material in transmission.

## 4. Conclusions

The influence of laser operating regimes on the bulk response of two types of glass was investigated. This study explored the effects of various processing parameters such as the burst repetition rate, scanning speed, and energy with the aim of establishing process windows for welding across three different regimes. The results revealed that the single-pulse regime is more suited for generating modifications with a teardrop shape without inducing cracks and exhibits a process window allowing for a pulse energy variation of more than two orders of magnitude. Additionally, high repetition rates help maintain this teardrop shape at higher scanning speeds. In contrast, the MHz burst regime tends to produce more defects, such as voids and cracks. This behavior is even more pronounced in the GHz burst regime, where the nonlinear absorption is extremely localized in the focal point; the lower energy per pulse only enables absorption near the focal point, leading to crack formation. Both sodalime and fused silica responded similarly to the different regimes, though voids were consistently generated in fused silica.

In conclusion, this work provides a comprehensive understanding of glass behavior under laser irradiation, enabling us to select appropriate parameters for welding applications.

## Figures and Tables

**Figure 1 materials-18-00078-f001:**
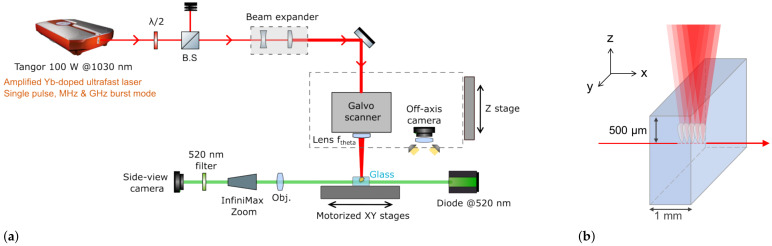
(**a**) Blueprint of the experimental setup used to produce bulk modifications in glass. (**b**) Representation of the laser beam passing inside the glass sample, generating modifications. The red arrow indicates the scanning direction.

**Figure 2 materials-18-00078-f002:**
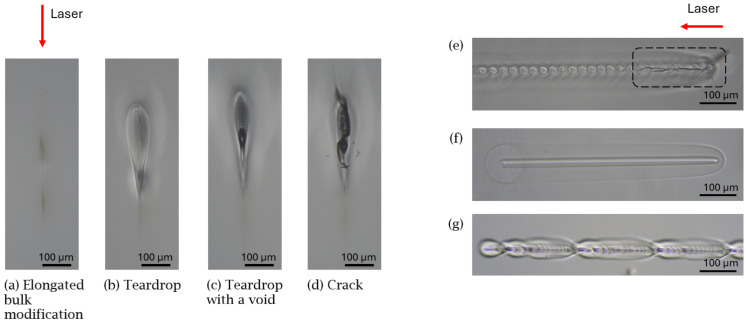
(**a**–**d**) Microscope side-view images of the different types of modifications produced in sodalime. Microscope top-view images of different bulk modifications: (**e**) crack at departure, (**f**) a stable seam, and (**g**) an unstable seam.

**Figure 3 materials-18-00078-f003:**
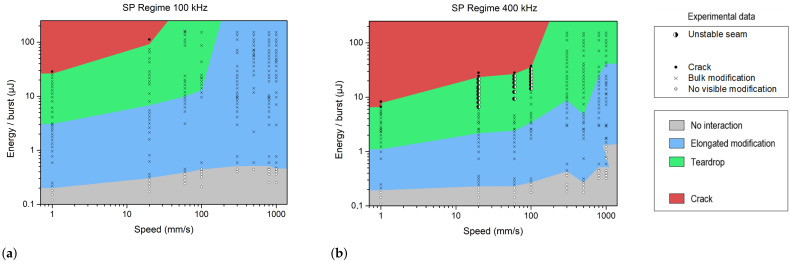
Schematic representation of the different operating windows observed during the single pulse study at a repetition rate of 100 kHz (**a**) and 400 kHz (**b**). The burst energy and speed values correspond to experimental data in sodalime. The different colors represent the various types of modifications.

**Figure 4 materials-18-00078-f004:**
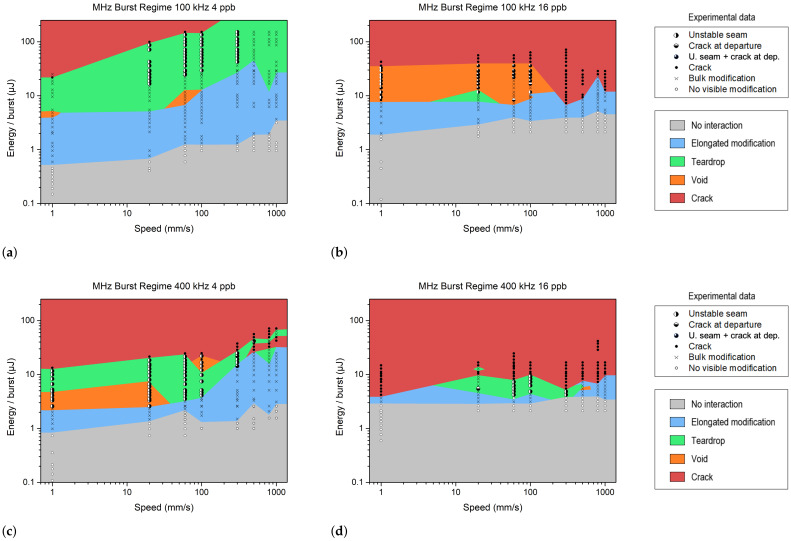
Schematic representation of the different operating windows observed during the MHz burst study. The burst energy and speed values correspond to experimental data obtained with sodalime. Bursts of 4 and 16 pulses were tested at 100 kHz (**a**,**b**) and 400 kHz (**c**,**d**), respectively. The various colors represent different types of modifications.

**Figure 5 materials-18-00078-f005:**
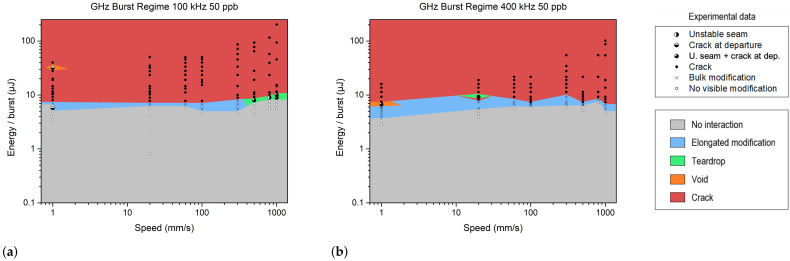
Schematic representation of the different operating windows observed during the GHz burst study at 100 kHz (**a**) and 400 kHz (**b**). The burst energy and speed values correspond to experimental data with sodalime. The different colors represent the various types of modifications.

**Figure 6 materials-18-00078-f006:**
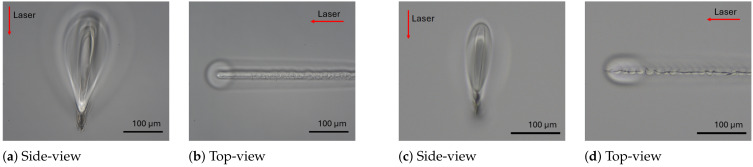
Microscope images showing a modification considered as cracks, with parameters of (**a**,**b**) 1 mm/s, 28 μJ/burst, 100 kHz, and GHz 50 ppb; (**c**,**d**) 20 mm/s, 9.5 μJ/burst, 400 kHz, and GHz 50 ppb.

**Figure 7 materials-18-00078-f007:**
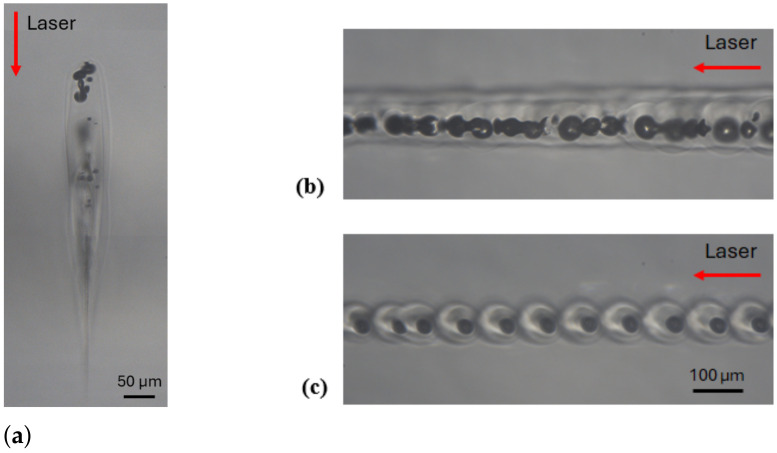
Images of modifications produced in fused silica: (**a**) teardrop shape (side view); (**b**) welding seam generated at different conditions; parameters were GHz burst 50 ppb, 400 kHz with 16 μJ/burst at 1 mm/s and (**c**) at 20 mm/s (top view).

**Figure 8 materials-18-00078-f008:**
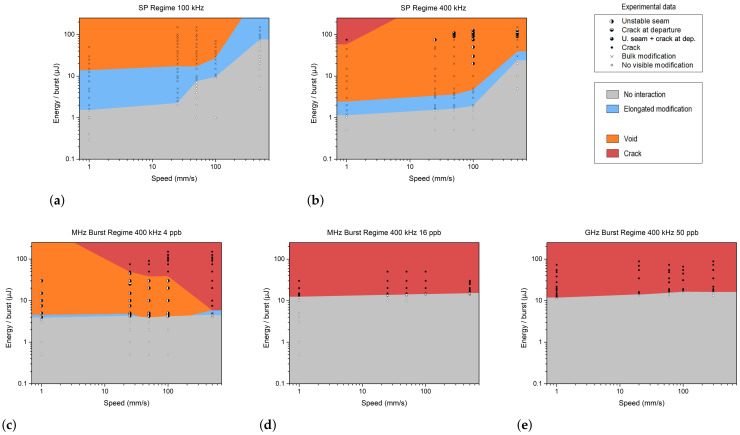
Schematic representation of the different operating windows observed in the single-pulse (**a**,**b**), MHz burst (**c**,**d**), and GHz burst (**e**) regimes. The burst energy and speed values correspond to experimental data obtained with fused silica. The different colors represent the various types of modifications.

**Figure 9 materials-18-00078-f009:**
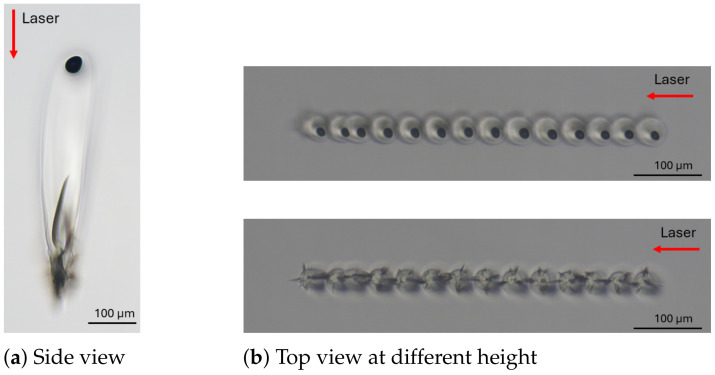
Images of modifications produced in fused silica: (**a**) a teardrop shape (side view); (**b**) the corresponding welding seam generated at 400 kHz with 16 μJ/burst at 20 mm/s in GHz burst with 50 ppb (top view at different heights).

**Figure 10 materials-18-00078-f010:**
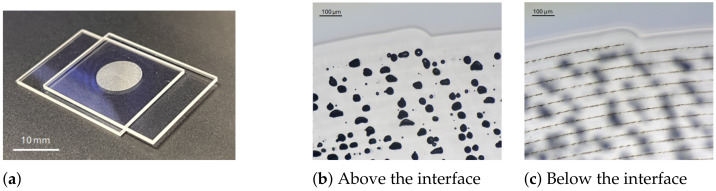
Fused silica samples welded with parameters selected from the processing window: (**a**) a photograph of the final assembly and (**b**,**c**) microscope images above (z = 200 μm) and below the interface (z = −200 μm).

**Table 1 materials-18-00078-t001:** Comparison of the ambient thermal properties of fused silica [32,33] and sodalime [34,35].

Material	Bandgap (eV)	Softening Temperature (°C)	Thermal Conductivity (W·m^−1^ ·K^−1^)	Thermal Expansion Coefficient (10^−6^ K^−1^)
Fused silica	9	1800	1.4	∼0.5
Sodalime	3–4	700	0.9	∼9

## Data Availability

The original contributions presented in this study are included in the article. Further inquiries can be directed to the corresponding authors.

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
