# Peer review of "In-Volume Glass Modification Using a Femtosecond Laser: Comparison Between Repetitive Single-Pulse, MHz Burst, and GHz Burst Regimes"

_materials, 2024, doi:10.3390/ma18010078_

Round 1
Reviewer 1 Report
Comments and Suggestions for Authors
This paper presents an in-depth comparative study of in-volume glass modification using femtosecond laser bursts in single pulse, MHz-burst, and GHz-burst regimes. The results are promising for understanding glass welding processes and defect formation. However, there are some points that should be improved as follows:
1. The description of crack formation in the GHz-burst regime needs further elaboration.
2. In Figure 6, the top-view and side-view images should explicitly indicate the scanning direction to provide more context for the crack morphology.
3. The authors mention "poor robustness" of GHz-burst regimes in transparent welding. Consider providing quantitative data to support this observation.
4. The study identifies void formation in fused silica but not sodalime. It is suggested to include additional discussion on the differences.
5. It is recommended to summarize key findings more concisely and include specific numerical data to strengthen the conclusions.
6. Minor grammatical errors and formatting inconsistencies are present throughout the text. A thorough proofread is recommended to improve clarity and readability.
Reviewer 2 Report
Comments and Suggestions for Authors
The submitted manuscript is entitled “In-Volume Glass Modification by Femtosecond Laser: Comparison Between Repetitive Single Pulse, MHz-Burst and GHz-Burst Regimes”.
In the presented study, the authors report on in-volume glass modifications produced by GHz-bursts of femtosecond pulses. Generally, the authors analyze the single pulse, MHz-burst, and GHz-burst regime methods of energy deposition in glass. The study focuses on establishing welding process windows for both sodalime and fused silica. The study of glass behavior under laser irradiation is of great importance in the case of various welding applications.
The manuscript reports interesting results to the materials science community. However, the authors should address the following minor issues:
1) The authors should discuss potential limitations in the modifications observed in the context of the application of the materials.
2) Line 179: The authors state that due to heat accumulation, a teardrop modification appears at lower burst energy compared to the repetitive single pulse regime and the minimum pulse energy to achieve teardrop formation is 1.2 μJ/pulse at 1 mm/s, 3.5 μJ/pulse at 100 mm/s and 43 μJ/pulse at 1 m/s.
The physical mechanism of the observed phenomena should be discussed in more detail.
3) Figure 1b) The quality of the figure is not sufficient. The scan direction should be drawn with a straight arrow.
Reviewer 3 Report
Comments and Suggestions for Authors
See the attachment for my feedback

Round 2
Reviewer 1 Report
Comments and Suggestions for Authors
My question has been addressed.